# Estrogen receptor α drives pro-resilient transcription in mouse models of depression

Zachary S. Lorsch [1], Yong-Hwee Eddie Loh[1], Immanuel Purushothaman[1], Deena M. Walker[1], Eric M. Parise[1], Marine Salery[1], Michael E. Cahill[2], Georgia E. Hodes[3], Madeline L. Pfau[1], Hope Kronman[1], Peter J. Hamilton[1], Orna Issler[1], Benoit Labonté[4], Ann E. Symonds[1], Matthew Zucker[1], Tie Yuan Zhang [5], Michael J. Meaney[5], Scott J. Russo [1], Li Shen [1], Rosemary C. Bagot[6] & Eric J. Nestler[1]

Most people exposed to stress do not develop depression. Animal models have shown that stress resilience is an active state that requires broad transcriptional adaptations, but how this homeostatic process is regulated remains poorly understood. In this study, we analyze upstream regulators of genes differentially expressed after chronic social defeat stress. We identify estrogen receptor α (ERα) as the top regulator of pro-resilient transcriptional changes in the nucleus accumbens (NAc), a key brain reward region implicated in depression. In accordance with these findings, nuclear ERα protein levels are altered by stress in male and female mice. Further, overexpression of ERα in the NAc promotes stress resilience in both sexes. Subsequent RNA-sequencing reveals that ERα overexpression in NAc reproduces the transcriptional signature of resilience in male, but not female, mice. These results indicate that NAc ERα is an important regulator of pro-resilient transcriptional changes, but with sex-specific downstream targets.

[1] Fishberg Department of Neuroscience, Friedman Brain Institute, Icahn School of Medicine at Mount Sinai, 1 Gustave L Levy Place, New York, NY 10029, USA. [2] Department of Comparative Biosciences, University of Wisconsin-Madison, Madison, WI 53706, USA. [3] School of Neuroscience, Virginia Polytechnic Institute and State University, 1981 Kraft Drive, Blacksburg, VA 24061, USA. [4] Department of Neuroscience and Psychiatry, Faculty of Medicine, Laval University, 2601 Chemin de la Canardière Québec, QC G1J 2G3, Canada. [5] Ludmer Centre for Neuroinformatics and Mental Health, Douglas Institute, Sackler Program for Epigenetics and Psychobiology, Departments of Psychiatry and Neurology and Neurosurgery, McGill University, 6875 Boulevard Lasalle, Montréal, QC H4H 1R3, Canada. [6] Departments of Psychology & Psychiatry, Ludmer Centre for Neuroinformatics and Mental Health, McGill University, 1205 Avenue Dr Penfield, Montréal, QC H3A 1B1, Canada. Correspondence and requests for materials should be addressed to E.J.N. (email: Eric.Nestler@mssm.edu)

Numerous studies have identified abundant transcriptional alterations in postmortem brains of patients with major depressive disorder (MDD)[1–7]. Given the inability to directly test causal relationships between gene expression levels and MDD in humans, animal models of depression-like behavior have been essential for determining the functional relevance of specific predicted regulatory genes. Stress paradigms in male and female mice including chronic social defeat stress (CSDS) and chronic variable stress (CVS) have been robustly validated for testing causality of genes regulating depression-like behavior[8–14]. However, recent RNA-sequencing and microarray studies in these paradigms report broad transcriptional changes[7,13–17] in the nucleus accumbens (NAc) and its afferent connections. Consequently, the contribution of individual causal genes to the wide-ranging transcriptional response is currently unknown.

Transcriptional changes have the potential to alter both cellular physiology and circuit-level signaling, which may have region-specific effects. For example, while activity in the ventral hippocampus to NAc projections promotes susceptibility, activity in prefrontal cortex (PFC) to NAc projections promotes resilience[18]. The causal relationship between transcriptional alterations, changes in cellular function, and modulated circuit-level physiology has been established by studies examining viral-mediated gene transfer in key driver genes known to regulate a single transcriptional network[7,16]. However, these studies have been limited to individual networks implicated in stress susceptibility. As such, there has to date been no comprehensive analysis of upstream genes that orchestrate the totality of these stress-induced transcriptional responses, particularly those that mediate homeostatic resilience to stress.

Prior work has shown that resilience is an active process that requires quantitatively greater transcriptional activity than stress susceptibility[9,16,17,19]. Being able to reproduce this resilient transcriptome in the brain of a patient with MDD has the potential to induce behavioral resilience and therefore may provide the basis for novel depression therapeutics. However, such approaches are dependent on a detailed understanding of resilient-specific transcriptional changes as well as their molecular regulators, which would be directly targeted with therapeutics. Studies that focus on bulk transcriptional changes involved in resilience are limited and the transcriptional drivers of such homeostatic resilience are currently unknown. In this study, we perform upstream regulator analysis on transcriptional datasets from male and female stress models. We find that *ESR1*, which encodes estrogen receptor α (ERα), is the top-predicted upstream regulator in male stress, but is regulated by stress in both sexes. We show further that overexpression of ERα increases behavioral resilience in both sexes, but the downstream transcriptional mechanism of ERα action in males and females is different.

## Results

**Regulation of ERα in NAc by chronic stress**. To identify transcriptional drivers of resilience to CSDS in an unbiased fashion, we used QIAGEN's Ingenuity® Pathway Analysis (IPA) on a publicly available RNA-sequencing dataset of differentially expressed genes in NAc of male mice 48 h after the last stress (Fig. 1a)[16]. The NAc is a central component of the brain's reward

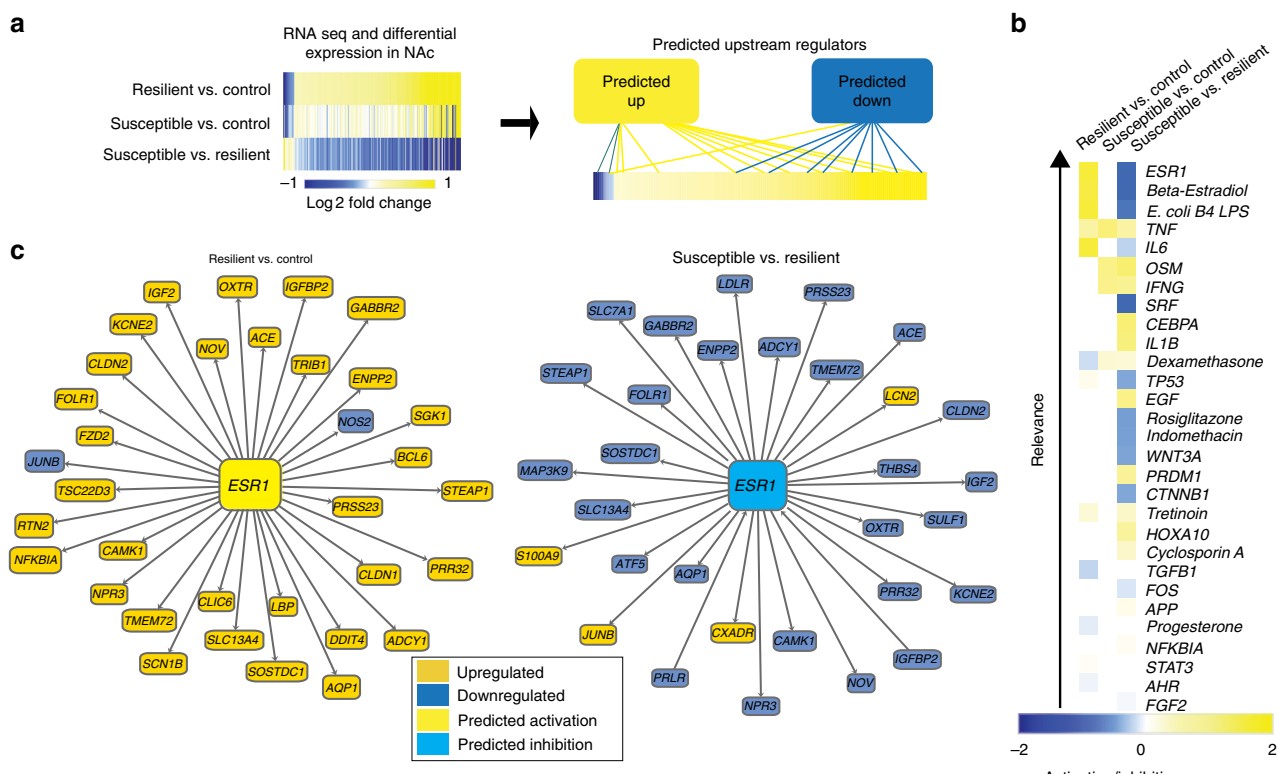

**Fig. 1** ERα is a predicted upstream regulator of pro-resilient transcriptional changes in NAc. **a** Experimental overview of upstream regulator analysis. NAc genes differentially expressed in control, CSDS susceptible, and CSDS resilient mice 48 h post defeat were used to predict upstream regulators. **b** Findings from IPA upstream regulator analysis on differentially expressed genes in CSDS. *ESR1*, which encodes ERα, is the top-predicted upstream regulator of differentially expressed genes (log fold change >1; $P < 0.05$) in the resilient vs. control (predicted activation; z-score = 3.16; $P < 6.64 \times 10^{-6}$) and susceptible vs. resilient (predicted inhibition; z-score = 3.22; $P < 1.38 \times 10^{-8}$) comparisons, with no predicted upstream regulator activity in susceptible vs. control. **c** Known interactions of differentially expressed genes (log fold change > 1) in the resilient vs. control and susceptible vs. resilient comparisons giving rise to the prediction of *ESR1* as an upstream regulator

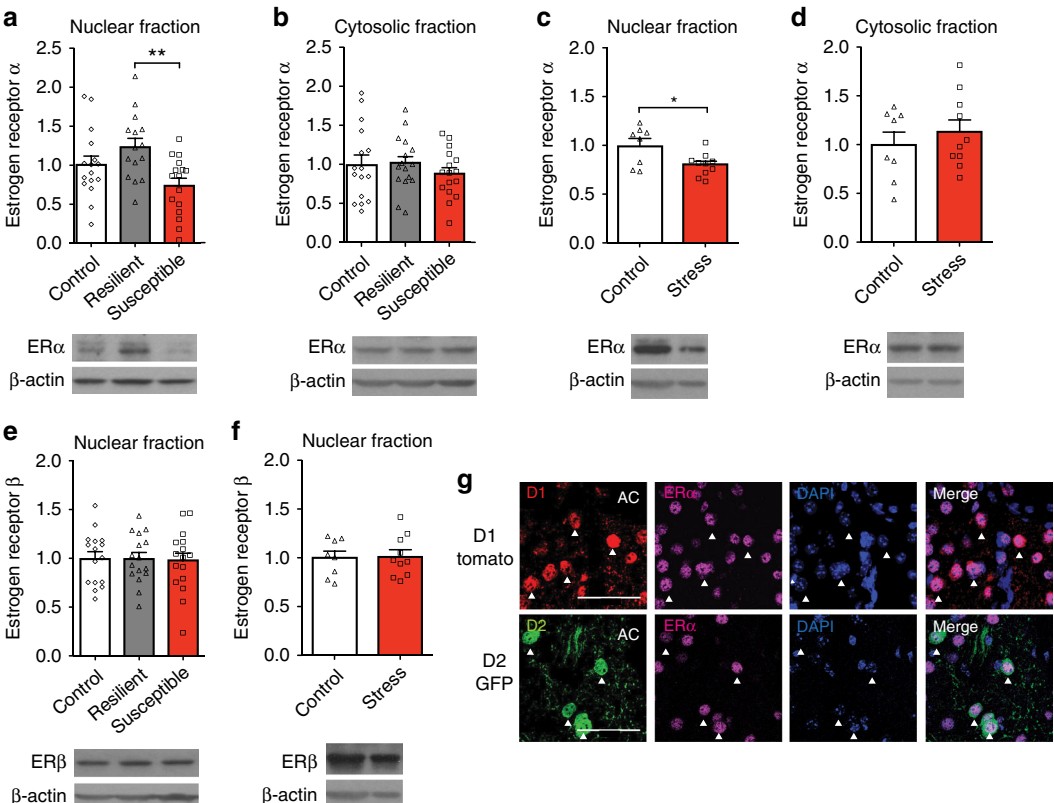

**Fig. 2** Stress regulation of ERα in the NAc of male and female mice. **a**, **b** In male mice, NAc ERα protein expression is increased in the nuclear ($F_{(2,44)}$ = 5.58, $P < 0.007$; $n = 16,15,16$) but not cytosolic ($F_{(2,44)}$ = 0.56, $P < 0.575$; $n = 16$) fractions of resilient ($P < 0.01$) mice after CSDS. **c**, **d** In female mice, NAc ERα protein levels are significantly decreased in the nuclear ($t_{(16)}$ = 2.67, $P < 0.017$; $n = 8,10$) but not cytosolic ($t_{(16)}$ = 0.76, $P < 0.456$; $n = 8,10$) fractions after CVS. **e**, **f** ERβ protein expression in the NAc nuclear fraction is not regulated by stress in males ($F_{(2,45)}$ = 0.05, $P < 0.954$; $n = 16$) or females ($t_{(16)}$ = 0.16, $P < 0.874$; $n = 8,10$). **g** ERα expression is enriched in the nucleus of both D1-expressing and D2-expressing medium spiny neurons in the NAc (AC = anterior commissure). Scale bars = 100 μM. Data are represented as mean ± SEM. **a**, **b**, **e** One-way ANOVA. **c**, **d** Two-tailed $t$test. *$P < 0.05$ $t$test, **$P < 0.01$ Tukey's post hoc test for multiple comparisons

circuitry involved in key features of depression-like behavior including anhedonia and social avoidance. This analysis, which predicts regulators from downstream transcriptional changes in known interaction partners, identified *ESR1* as the top-predicted upstream regulator of genes differentially expressed between resilient vs. control and resilient vs. susceptible mice, with no predicted regulation of differential expression between susceptible vs. control mice (Fig. 1b). As *ESR1* was predicted to be activated in NAc in the resilient phenotype (Fig. 1c), we hypothesized that ERα mediates pro-resilient gene transcription in this brain region.

In addition to NAc, the PFC has been implicated in stress resilience[18]. As such, we examined upstream regulators of CSDS-dependent transcriptional changes in this region (Supplementary Fig. 1a). Unlike in NAc, *ESR1* was not predicted to activate or inhibit transcriptional changes in PFC. Likewise, to determine whether *ESR1* was regulated in NAc of other stress models, we analyzed upstream regulators of CVS (Supplementary Fig. 1b). Although CVS does not produce an identifiable resilient subpopulation, we were able to examine the effects of stress in both male and female mice, which show sex-specific responses. While 6-day CVS is sufficient to reliably induce depression-like behavioral alterations in females, males do not show behavioral changes[10,15]. We found a predicted inhibition for *ESR1* in male CVS NAc with no prediction for females.

To investigate the role of ERα as a driver of resilience, we generated an independent cohort of male CSDS mice (Supplementary Fig. 2a, b) and examined changes in ERα protein expression. Consistent with our prediction, ERα levels were

increased in NAc nuclear fractions of resilient mice (Fig. 2a) with no significant differences in cytosolic fractions (Fig. 2b). Further, nuclear ERα expression positively correlated with social interaction (SI) score, indicating a direct association with resilient behavior (Supplementary Fig. 2c). Despite our lack of prediction for *ESR1* in female CVS (Supplementary Fig. 1b), we used this paradigm to test if ERα was regulated as in males. We identified decreased levels of ERα in nuclear (Fig. 2c) but not cytosolic (Fig. 2d) NAc fractions in stressed mice with no difference in estrous cycle phase between groups (Supplementary Fig. 2d). These changes were unique to ERα, since nuclear ERβ levels did not change after either CSDS or CVS (Fig. 2e, f). Similarly, neither stress paradigm affected cytosolic levels of ERβ or aromatase (Supplementary Fig. 2e–h), although there were baseline sex differences in ERα, ERβ, and aromatase expression (Supplementary Fig. 3).

Despite our finding of altered ERα in bulk tissue, it is possible that ERα localizes to a specific subset of neurons. To examine the potential cell-type specificity of nuclear ERα changes we performed ERα immunohistochemistry (IHC) in NAc of dopamine receptor 1 (D1) and dopamine receptor 2 (D2)-labeled transgenic mice. Consistent with our Western blot analysis, ERα was enriched in the cell nucleus, but we found consistent expression among all D1-expressing or D2-expressing cells visualized (Fig. 2g).

**ERα overexpression in NAc is pro-resilient in both sexes.** To directly investigate the ability of ERα to induce stress resilience,

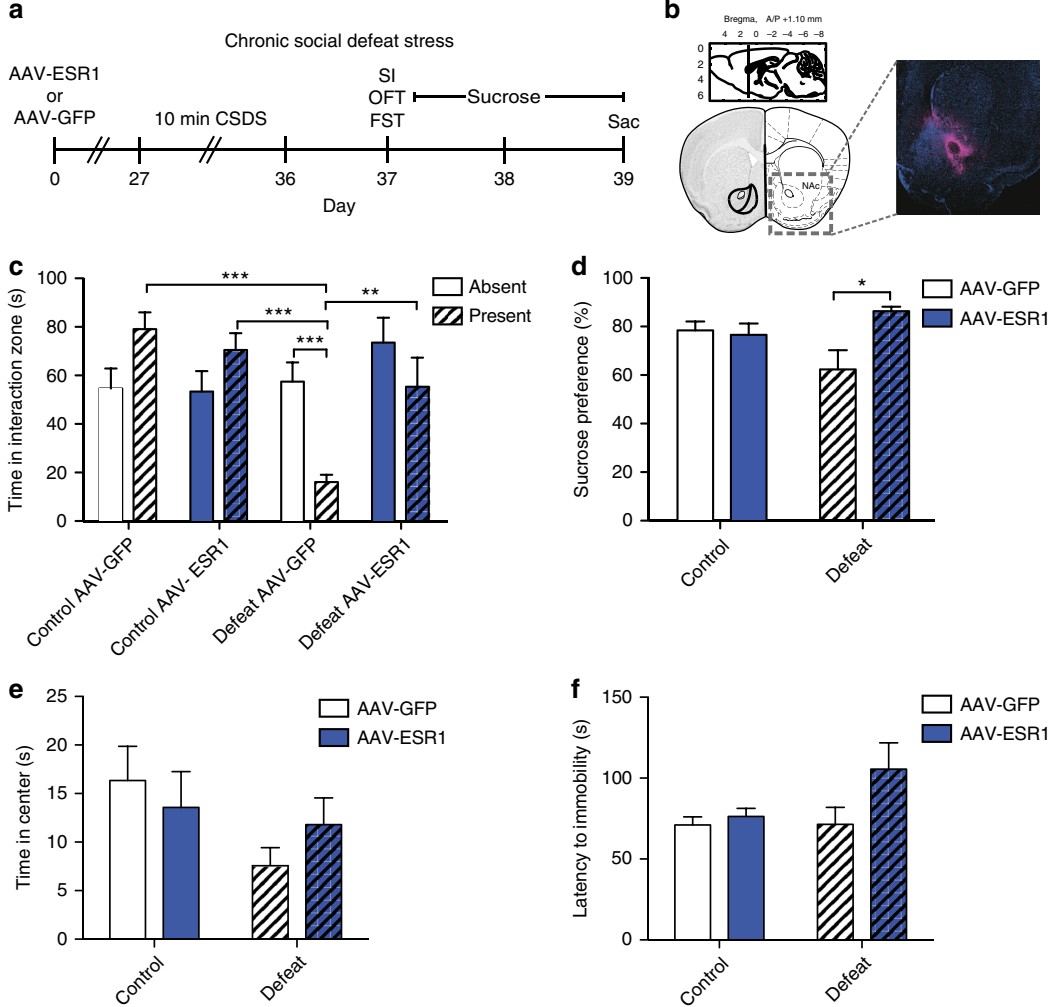

**Fig. 3** ERα overexpression in NAc induces antidepressant-like effects in CSDS in male mice. **a** Timeline for CSDS experiment. **b** Location of targeted site within the NAc for viral-mediated gene transfer and IHC validation of viral expression (magenta) against DAPI background (blue). **c** Male mice show significant ($\chi^2(7) = 23.67$, $P < 0.001$, $n = 10,8,8,9$) group differences in time spent in the interaction zone in the social interaction test. Defeated AAV-GFP mice, but not any other group, spend less time in the interaction zone when the CD1 aggressor is present than when the CD1 aggressor is absent ($P < 0.001$). Additionally defeated AAV-GFP mice spend less time in the interaction zone when the CD1 aggressor is present than control AAV-GFP mice ($P < 0.001$), control AAV-ESR1 mice ($P < 0.001$), and defeated AAV-ESR1 mice ($P < 0.01$). **d** There is a non-significant trend ($\chi^2(3) = 7.72$, $P < 0.052$, $n = 9,8,8,8$) towards altered sucrose preference in the sucrose preference test with defeated AAV-ESR1 mice consuming significantly ($P < 0.05$) more sucrose than defeated AAV-GFP mice. **e, f** CSDS ($F_{(1,32)} = 2.96$, $P < 0.095$; $n = 10,8,9,9$) and ERα overexpression ($F_{(1,32)} = 0.05$, $P < 0.822$) do not have an effect on time spent in the center in the open field test or latency to immobility ($\chi^2(3) = 5.76$, $P < 0.124$, $n = 9,8,7,9$) in the forced swim test. Data are represented as mean ± SEM. **c**, **d**, **f** Kruskal–Wallis test. **e** One-way ANOVA. *$P < 0.05$, **$P < 0.01$, ***$P < 0.001$ post hoc independent-samples Mann–Whitney

we injected adeno-associated virus (AAV)-ESR1 or AAV-GFP into NAc and exposed male mice to CSDS (Fig. 3a, b). While defeated mice injected with AAV-GFP spent less time in the interaction zone when the CD1 aggressor was present (Fig. 3c), all other groups, including defeated mice injected with AAV-ESR1, did not. Moreover, control mice, as well as defeated AAV-ESR1 injected mice, spent more time in the interaction zone when the CD1 was present than defeated AAV-GFP mice. Defeated ERα-overexpressing mice also displayed an increased sucrose preference over that of defeated GFP mice (Fig. 3d), with no effect of virus in non-defeated controls. There was no effect of CSDS or virus on time spent in the center in the open field test (OFT) (Fig. 3e) or latency to immobility in the forced swim test (FST) (Fig. 3f).

Importantly, injection of AAV-ESR1 produced an increase of *ESR1* in the cell nucleus of both D1-expressing and D2-expressing expressing neurons (Supplementary Fig. 4). Additionally, CSDS

or virus did not produce changes in overall locomotion (Supplementary Fig. 5a), and there was no effect of virus on body weight at the time of behavior (Supplementary Fig. 5b). To determine if, despite a null prediction (Supplementary Fig. 1a), ERα overexpression in PFC could increase resilience, we injected AAV-ESR1 or AAV-GFP into PFC and exposed mice to CSDS. Both AAV-GFP and AAV-ESR1 injected mice spent less time in the interaction zone when the target was present, but unlike in NAc, there were no differences between virus groups (Supplemental Fig. 5c). Further, ERα overexpression did not affect sucrose preference in defeated mice (Supplemental Fig. 5d).

To assess whether ERα overexpression regulates pro-resilient behavior in female mice, we injected AAV-ESR1 or AAV-GFP into NAc and exposed mice to 6-day CVS (Fig. 4a). In order to evaluate differences in overall behavior resulting from stress and ERα overexpression, we performed exploratory factor analysis on the totality of the female behavior. Factor 1 scores, which explain

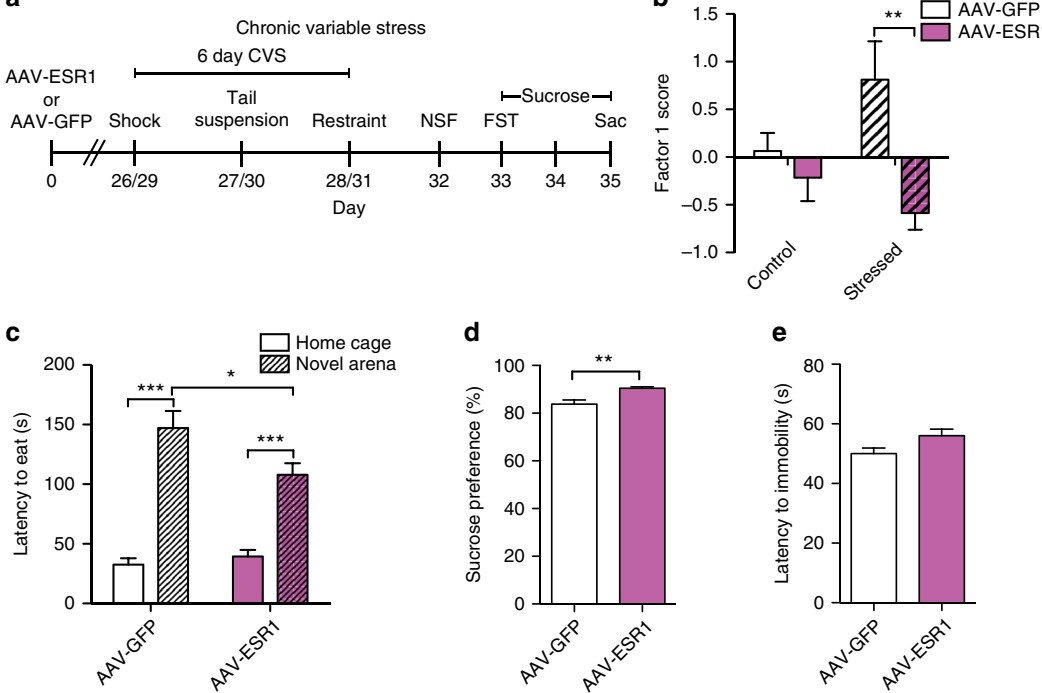

**Fig. 4** ERα overexpression in NAc induces antidepressant-like effects in CVS in female mice. **a** Timeline for CVS experiment. **b** Virus ($F_{(1,42)} = 10.06$, $P < 0.003$; $n = 11,12,11,12$), but not stress ($F_{(1,42)} = 0.52$, $P < 0.478$) has a significant main effect on factor 1 scores. However, there is a significant stress by virus interaction ($F_{(1,42)} = 4.47$, $P < 0.040$) whereby stressed, but not control, AAV-GFP injected mice have higher factor 1 scores ($P < 0.01$). **c** Mice exposed to CVS take longer to feed in a novel environment than the home cage in the novelty suppressed feeding test ($F_{(1,20)} = 116.13$, $P < 0.001$; $n = 11$). However, this effect is mediated by virus as there is a significant virus by arena interaction ($F_{(1,20)} = 7.73$, $P < 0.014$) with AAV-GFP injected mice showing a longer latency to feed in the novel environment than AAV-ESR1 injected mice ($P < 0.05$). **c, d** Stressed AAV-ESR1 mice consume more sucrose ($U = 19.0$, $P < 0.007$, $n = 11$) and show a non-significant trend towards a higher latency to immobility ($t_{(19)} = 2.03$, $P < 0.053$; $n = 10,11$) in the forced swim test than stressed AAV-GFP mice. Data are represented as mean ± SEM. **b** Two-way ANOVA. **c** Mixed-model ANOVA. **d** Independent-samples Mann–Whitney. **e** Two-tailed $t$ test. $*P < 0.05$ in **c**, $**P < 0.01$, $***P < 0.001$ Bonferroni post hoc for independent samples

the most variance in the female behavioral data (Supplemental Fig. 6a, b), were dependent on the combination of exposure to CVS and virus, with stressed AAV-GFP mice exhibiting higher factor 1 scores than stressed AAV-ESR1 mice. In terms of individual behaviors, NAc ERα overexpression reduced latency to feed in a novel environment in the novelty suppressed feeding (NSF) test in stressed females (Fig. 4c), an antidepressant-like effect[15]. Stressed ERα-overexpressing mice also showed increased sucrose preference (Fig. 4d) and a non-significant trend towards increased latency to immobility in the FST (Fig. 4e) relative to stressed GFP controls. In unstressed controls, ERα overexpression reduced latency to eat in the home cage in NSF, had no effect on sucrose preference, and increased latency to immobility in FST (Supplementary Fig. 6c–e)

**Sex-specific transcriptional effects of ERα.** To test our hypothesis that ERα drives the transcriptional changes seen in NAc of resilient mice, we quantitatively assessed the similarity of transcriptional profiles using rank-rank hypergeometric overlap (RRHO)[20] to determine if ERα overexpression recapitulates a resilient transcriptional profile (Fig. 5a). Importantly, since our behavioral data indicated that ERα overexpression in NAc was protective against future stressors, we analyzed the transcriptional effects of ERα in the absence of stress. In NAc, genes most strongly upregulated by overexpression of ERα at baseline showed a dramatic (maximum –log $P$ value = 30.34) overlap with the genes most strongly upregulated in resilience after CSDS (Fig. 5b). In contrast, negligible overlap was seen

with the susceptible phenotype (maximum –log $P$ value = 9.80) (Fig. 5c). We repeated this analysis using a different CSDS dataset sequenced 30 days after the last stress exposure[17] and found even stronger (maximum –log $P$ value = 305.36) overlap between ERα overexpression and the resilient vs. susceptible comparison amongst upregulated genes, with again no overlap between ERα overexpression and susceptible vs. control conditions (maximum –log $P$value = 4.62). To categorize these downstream effectors in terms of biological function, we extracted 613 co-upregulated genes in the most significant RRHO overlap gene set and performed Gene Ontology (GO) analysis. We found statistical overrepresentation of nervous system and other developmental and cellular processes as well as an underrepresentation of sensory response genes (Fig. 5d) amongst this gene set.

Unlike CSDS, CVS (6 days) does not generate a clear resilient phenotype in females, and males are unaffected by this paradigm[10,15]. As such, we did not expect to recapitulate the overlap between gene expression changes induced by ERα overexpression and CVS exposure in females, as seen for ERα overexpression and CSDS resilience in males. However, given that ERα levels are decreased in NAc of CVS females, and ERα overexpression renders females resilient to CVS, we tested whether ERα overexpression might induce transcriptional changes opposite to those induced by stress. Contrary to this possibility, ERα overexpression showed no specific pattern of overlap with the CVS stress phenotype in female mice (maximum –log $P$value = 14.73) (Fig. 5e).

Since our transcriptional signatures for male and female mice were acquired under baseline conditions, we could compare the

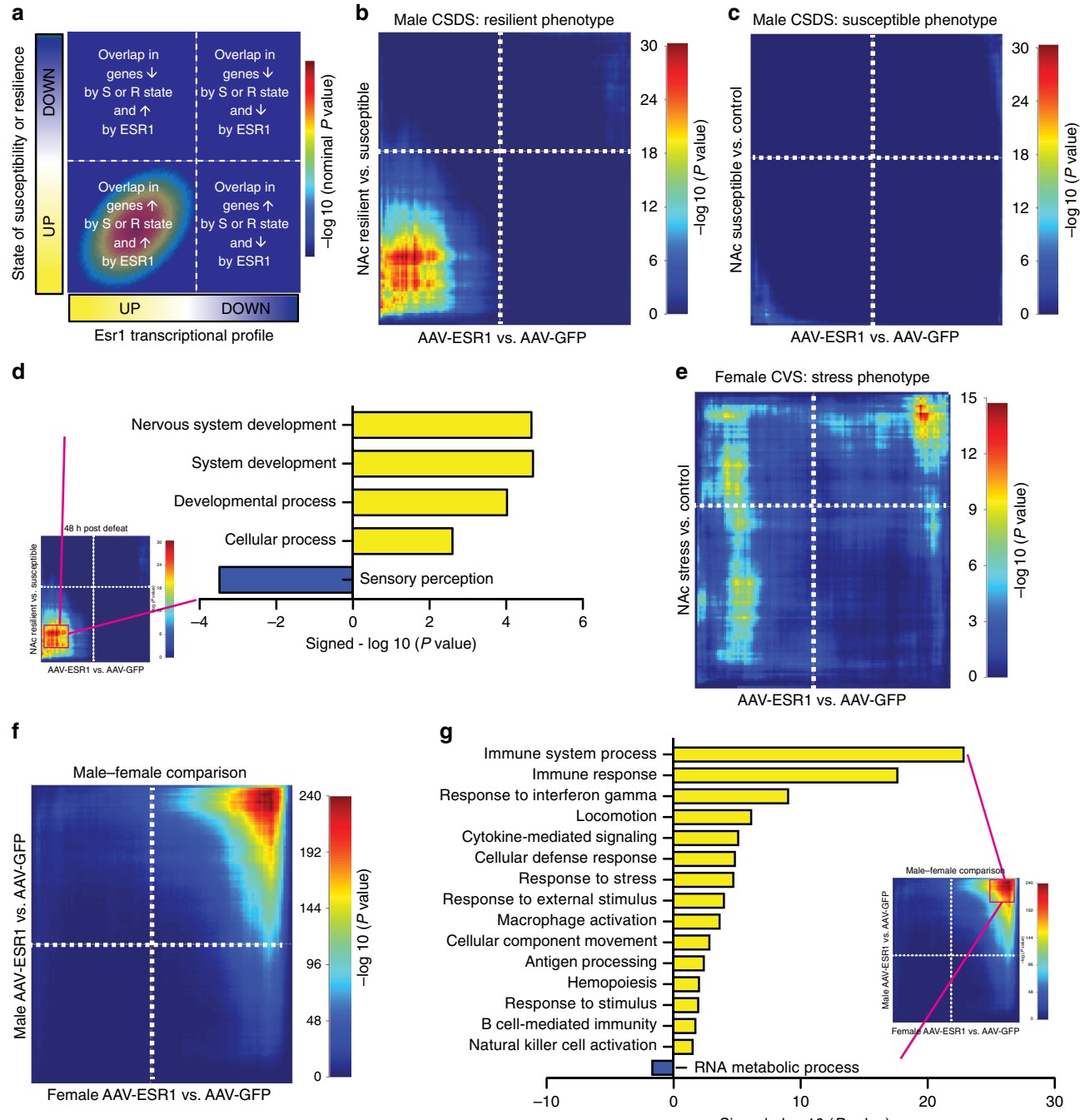

**Fig. 5** ERα overexpression in NAc recapitulates the transcriptional signature of resilience in male mice. **a** Overview of RRHO. **b** The genes most strongly upregulated by ERα overexpression in the male NAc overlap with genes most strongly upregulated in this region of resilient mice after CSDS. **c** The male transcriptional signature of ERα overexpression is distinct from the transcriptional signature of CSDS susceptibility. **d** GO analysis of transcripts co-upregulated by ERα overexpression and the resilient phenotype after CSDS shows a positive enrichment for genes implicated in nervous system development and a negative enrichment for sensory perception genes. **e** ERα overexpression in the NAc of female mice induces a transcriptional pattern distinct from that induced by CVS. **f** ERα overexpression in male and female mice shows an overlap only in the most strongly downregulated genes. Genes upregulated by ERα overexpression, or moderately regulated in either direction by ERα, do not overlap. **g** GO analysis of transcripts co-downregulated by ERα in males and females indicates positive enrichment with immune-associated functions and underrepresentation of RNA metabolic processes

two directly. As such, we next compared the effects of ERα overexpression in NAc of male vs. female mice to determine whether the downstream transcriptional actions are conserved. While there was robust coexpression amongst the most down-regulated genes (maximum −log $P$ value = 239.52), the upregulated genes, which were the strongest area of overlap between male ERα overexpression and resilience (Fig. 5b), did not overlap

significantly (Fig. 5f). To determine the biological functions associated with the shared overlap in downregulated genes, we performed GO analysis on the 320 most significant RRHO-identified genes. Within this set, we found statistical over-representation of immune-related processes with an under-representation of terms related to RNA metabolism (Fig. 5g).

## Discussion

Clinically, numerous associations between estrogen signaling and MDD have been reported. For example, women are more likely to suffer from MDD than men[21], and periods during which estrogen levels are changing, such as postpartum and perimenopause, are associated with higher rates of depression[22,23]. In addition, some data suggest that use of hormonal contraception is associated with a higher risk of depression[24], especially during adolescence when certain contraceptives decrease estrogen levels by suppressing the hypothalamic-pituitary-ovarian axis[25]. Despite these links, the mechanisms underlying estrogen action have remained elusive. Further, the role of estrogens in males, who have both circulating estrogens and the ability to convert androgens to estrogens locally in the brain via aromatase[26], has been minimally explored.

Our findings implicate ERα as a master regulator of transcriptional changes that drive resilience to CSDS in males (Fig. 1). While genetic alterations in ERα have been linked to MDD in humans in candidate gene studies[27,28], these findings are unique to females[28]. Surprisingly, our upstream regulator analysis in CVS identified *ESR1* in male, but not female, stress (Supplementary Fig. 1b). Although the direction of the prediction in males (inhibition by stress) is consistent with a pro-resilient role for ERα, male mice (unlike female mice) do not show behavioral alterations following the 6-day CVS paradigm[15]. Despite this finding, ERα levels were decreased by CVS in the female NAc nuclear fraction (Fig. 2c) and ERα overexpression increased resilience in CSDS (Fig. 4). Consequently, our results indicate that ERα acting in NAc is pro-resilient in both sexes. Since *ESR1* was not a predicted regulator in PFC (Supplementary Fig. 1a) and overexpression of ERα in PFC had no effect in CSDS (Supplementary Fig. 5c, d), our data indicate that this effect is region-specific.

While there is some evidence for a possible antidepressant effect of ERα, for example, injections of an ERα agonist have been shown to mitigate postpartum depression-like responses in rats with resulting changes in *BDNF* and pERK signaling[29], our study identifies transcriptional regulation of numerous downstream targets (Fig. 5) as the basis of ERα action. It is well established that estrogen can contribute to diverse effects within the male and female brain[30], and some effects of estrogens, such as the rapid induction of plasticity in the dopaminergic system[31,32], are likely a product of membrane-bound ERs that are more prominent in NAc[33]. However, even within these areas, some effects of estrogen occur on a time scale indicative of transcriptional action[34]. Our findings that ERα is modulated by stress paradigms only in the nuclear fraction (Fig. 2a–d), is enriched in the nucleus of D1-expressing and D2-expressing neurons (Fig. 2g), and mimics the transcriptional effects of resilience (Fig. 5b) suggest that the nuclear actions of ERα mediate its antidepressant effects.

Importantly, while resilient mice show higher levels of ERα in NAc nuclear fractions than susceptible mice, control mice show an intermediate phenotype (Fig. 2a) with nuclear ERα levels directly correlating with resilient behavior (Supplementary Fig. 2c). Further, female CVS, which does not include a clearly defined resilient population, decreases nuclear ERα levels (Fig. 2c). These findings suggest that ERα expression in NAc decreases from baseline in the presence of stress, but increases from baseline as part of resilience. As such, alterations of ERα have the potential to change the transcriptional profile whereby the downstream targets of ERα would be most affected in resilience and least affected in susceptibility.

Our results indicate that ERα can induce the transcriptional signature of resilience in the absence of stress (Fig. 5b). While it has been reported that antidepressants such as ketamine and imipramine can induce resilience-specific gene expression

profiles[17], this only occurs in drug responders. Our findings, in contrast, suggest that ERα could be used to boost active resilience mechanisms prior to stress. Previous studies in animal models have indicated that neural estrogens can protect against the detrimental cognitive effects of stress in both females and males[35]. Our data extend these findings by reporting that ERα specifically (Fig. 1), as well as its downstream transcriptional effects (Fig. 5), are essential to this mechanism and the resulting protective effects.

The NAc contains two principal types of medium spiny neurons with different functional properties. Although ERα is expressed in both D1-expressing and D2-expressing neurons (Fig. 2g) and our AAV-ESR1 virus shows tropism for both cell types (Supplementary Fig. 4), cell-type-specific contributions to both pro-resilient behavior and ERα-dependent transcriptional profiles are possible and warrant future study.

Our upstream regulator analysis is the first in silico approach to identify drivers of CSDS transcriptional changes. This marks a significant departure from previous in vivo studies[8–12,15,19], which selected targets based on a priori hypotheses related to known alterations in brain function. Our approach was successful in that it identified both upstream regulators previously implicated in MDD, such as interleukin-6[36–38], tumor necrosis factor[37], and β-catenin (*CTNNB1*)[9], as well as novel genes such as *ESR1*, which was the top regulator (Fig. 1b). Further, our data suggest that while both CSDS and CVS uniquely affect NAc ERα signaling (Fig. 2a–d), and increased NAc ERα signaling is pro-resilient in both paradigms (Figs 3–4), the relevant downstream transcriptional effects of ERα are sex-specific. This is consistent with RNA-sequencing studies demonstrating divergent stress-dependent transcriptional responses in NAc of male vs. female mice[7,15] as well as research demonstrating that estrogens can produce identical effects in males and females through different mechanisms[39]. Although there was considerable overlap in the genes downregulated by ERα in NAc of male vs. female mice (Fig. 5f), the genes upregulated by ERα, which overlapped with the resilient phenotype in male CSDS (Fig. 5b), were not apparent in females. However, our finding that ERα downregulates immune response genes in the male and female NAc (Fig. 5g), and the established associations between inflammation and MDD[36–38], suggests that this process may be a universal mechanism that also contributes to the antidepressant potential of ERα. Despite consistency between males and females in the directionality of changes in estrogen-related proteins in NAc (Fig. 2), we found baseline sex differences across nuclear and cytosolic fractions (Supplementary Fig. 3) that could contribute to altered ERα-dependent transcriptional profiles. Since recent studies have modified the CSDS protocol to successfully induce depression-like behavior in female C57BL/6J mice[40,41], future studies can take advantage of this paradigm to further explore these sex differences and determine whether, similar to males, the transcriptional activation induced by ERα in the female NAc recapitulates the transcriptional profile of female resilience.

Our findings reveal that ERα signaling in NAc drives much of the transcriptional changes necessary for homeostatic resilience[16,17,19] and that, even in the absence of stress, ERα can induce a resilient-like transcriptional state. This extends to both male and female stress paradigms, although the downstream transcriptional effects of ERα are sex-specific for upregulated, but not downregulated, genes. Together, these results provide critical mechanistic insight into the regulation of resilience that may be useful for the development of novel antidepressants.

## Methods

**Upstream regulator analysis.** Upstream regulator analysis was performed using the upstream regulator tool in QIAGEN's Ingenuity® Pathway Analysis (Qiagen

Redwood City, CA, USA, www.qiagen.com/ingenuity). This function predicts the identity and direction of change of upstream regulators for a given differential expression signature from the magnitude and scale of gene expression changes in a dataset. Predictions used in this study were based on experimentally observed interactions within all datasets in IPA with the stringent filter setting applied. Reported $P$ value calculations were determined from the Ingenuity Knowledge Base reference set considering both direct and indirect relationships. Interactions from all nodes, data sources, species, and mutations were utilized to predict regulators, but data from tissues and cell lines were excluded. In order to provide a high confidence for regulators, bioinformatically predicted interactions that have not yet been shown experimentally were excluded. Any regulator for which there was sufficient evidence to generate an activation/inhibition prediction is reported. Input to IPA was prepared from our previous work describing transcriptional changes by RNA-sequencing in the NAc and PFC 48 h after CSDS (Gene Expression Omnibus database: GSE72343)[16] and the NAc 48 h after CVS (6 days) (Gene Expression Omnibus database: GSE85136)[15]. Data were filtered for protein-coding genes. Three separate comparisons (susceptible vs. resilient, resilient vs. control, and susceptible vs. control) were utilized for CSDS, and one comparison (stressed mice vs. unstressed mice) was used for male and female mice in CVS. All sequenced genes from these datasets were uploaded to IPA along with expression (log fold change) and significance ($P$ value) values for every differential expression comparison. In order to identify top upstream regulators, IPA upstream regulator analysis was first performed on a limited dataset including only the most differentially expressed genes (log fold change > |1| and an expression $P$ value < 0.05). Statistically significant ($P$ value < 0.05) upstream regulators in each dataset were combined in a region-specific comparison analysis to visualize common regulators between differential expression comparisons. Regulators were sorted by activation $z$-score to reveal the strongest predicted regulators across a dataset. Notably, while IPA identifies multiple regulators as statistically significant ($P$ value < 0.05), only regulators generating a prediction for activation ($z$-score > 0) or inhibition ($z$-score < 0) were analyzed. Once upstream regulators were identified, all possible associations with a known regulator were defined by lowering the filter cutoff to only log fold change >|1|.

**Animals and behavior**. Mice were maintained on a 12 h light–dark cycle with lights on at 7:00 a.m. and a controlled temperature range of 22–25 °C. Food and water were provided ad libitum except for the 24 h preceding NSF testing, when food was removed. All experiments conformed to the Institutional Animal Care and Use Committee guidelines at Mount Sinai. All behavioral testing was counterbalanced across experimental groups, and assignment to experimental groups was random. Behavioral analysis was performed either automatically by video tracking software (Ethovision 10.0, Noldus), manually on pre-recorded video by investigators blind to study design, or manually in real time. To ensure adequate power, sample sizes were chosen in accordance with number of mice needed to show statistical significance in CSDS and CVS as defined by previous studies[15,16]. Experiments were not replicated unless otherwise specified.

CSDS and SI tests were performed according to our established protocol[8,19]. Eight-week-old male C57BL/6J mice (Jackson Labs) were subjected to ten daily 10-min defeats by a novel CD1 retired breeder aggressor mouse (Charles River Laboratories) that had been previously screened for aggressive behavior. Screening took place in 3-min intervals over the 3 days preceding the first defeat. Mice that attacked C57Bl/6J screener mice on all 3 days were selected for inclusion in the study. Aggressive CD1 mice were placed on one side of a large hamster cage separated by a perforated Plexiglas divider. At the onset of the CSDS, a single C57BL/6J mouse was placed into the same side of the cage as the CD1, and the CD1 was given 10 min to physically attack the C57BL/6J mouse. At the end of these 10 min, the C57BL/6J mouse was moved to the other side of the Plexiglass divider, where there was no more physical contact. However, as the divider was perforated, sensory contact continued. After 24 h of sensory contact, the C57BL/6J mouse was moved into the adjacent hamster cage where the procedure was repeated. Control mice were housed in a mouse cage with a Plexiglas divider. To simulate the CSDS protocol without the physical or emotional stress, another C57BL/6J mouse was placed on the opposite side of the divider, but no physical contact was allowed between the two mice. In order to control for experimental handling effects, control C57BL/6J mouse were moved to a new half of a cage every day during the defeat procedure. CSDS was completed in 10 days for all experiments with the exception of AAV-ESR1/AAV-GFP injection into the PFC, whereby 10 days was insufficient to establish a baseline effect in the SI test and an additional 2 days of defeat were performed. After the final defeat, C57BL/6J mice were single-housed in preparation for behavioral testing the subsequent day.

All behavioral tests took place in a behavior suite different from where defeat was performed. Behavior testing took place during the animals' light cycle. Mice were given 1 h to habituate to the room prior to behavioral testing. In all cases of CSDS, SI testing was the first behavioral test performed. SI testing was performed under red light in a closed behavioral chamber. C57BL/6J mice were placed into an open arena with an empty cage at one side (interaction zone). Mice were given 2.5 min to explore the arena and then removed. A novel CD1 aggressor to which the C57BL/6J mouse had never been exposed to was placed in the cage (interaction zone) and the procedure was repeated. Time in the interaction zone and locomotion were recorded automatically with video tracking software. Data were

analyzed as time spent in the interaction zone when the aggressor was absent compared to time spent in the interaction zone when the aggressor was present. SI ratio was calculated as (time in the interaction zone with a target mouse present)/ (time in the interaction zone with target absent). For protein analysis, defeated mice with SI ratio scores >1.2 that spent >60 s in the interaction zone when the target was present were determined to be resilient. Defeated mice with SI ratio scores <0.8 that spent <40 s in the interaction zone when the target was present were determined to be susceptible. In addition, locomotion was assessed as total distance moved when the aggressor was absent.

OFT was performed on the same day as social interaction. To minimize spillover effects of subsequent behavioral tests, mice were given 2 h in the home cage prior to the next behavioral test. During OFT, mice were given 10 min to explore an open arena under red light. Although nothing was physically placed in the arena, center and periphery were defined in video tracking software. Total time in the center was recorded and utilized for analysis.

FST was also performed on the same day as social interaction with an additional 2 h in the home cage prior to testing to minimize possible spillover effects. Since FST is the most stressful test, it was always performed last. FST was performed in 4 L Pyrex beakers filled with two liters of 25 °C (±1°) water. Mice were placed into the water and recorded by a front-facing camera for a period of 6 min. Investigators blind to study design scored FST videos by dividing the 6 min segments into 72 five-second intervals and recording latency to immobility swimming or floating behavior during each of those intervals.

CVS was performed as previously described[15] with three different hour-long stresses performed twice over a total of 6 days. Females utilized for CVS were 8-week-old female C57BL/6J mice. The order of stressors was foot shock (0.45 mA) on days 1 and 4, tail suspension on days 2 and 5, and restraint stress (in a 50 mL Falcon tube in the home cage) on days 3 and 6. Mice were group housed (five mice per cage) when they were not being stressed and control mice remained in their home cages throughout. After the last stressor, stressed and unstressed control mice were single-housed in preparation for behavior the following day. Due to logistical reasons, behavior on CVS stressed mice and unstressed controls were staggered by 1 day. Accordingly, direct comparisons between these groups could not be performed. However, the sequence of behavioral tests was identical for each group.

In order to determine estrous cycle phase, female mice were smeared vaginally according to previously described protocols[15]. Estrous phase on the day of sacrifice was used for analysis, and mice in the estrous stage were excluded from the analysis. Behavior on CVS mice included NSF, FST, and sucrose preference tests, which occurred on subsequent days. These tests were adapted from published studies[10,15,16,19] with specific methodological details and any deviations described herein.

NSF was performed on the first post-stress day. On the last day of stress, mice were single-housed in a new cage without food to undergo 24 h of food deprivation. The following day, mice were placed in a novel arena with corncob bedding and a single piece of food in the center of the arena. Time to feed was recorded manually under white light. Mice were given a maximum of 10 min to eat, after which the trial was ended and latency of 600 s was recorded. After the mouse ate in the novel arena, the mouse was returned to the home cage where a single piece of food was located in the center, and time to eat in the home cage was recorded. Data were analyzed as latency to eat in the novel arena and latency to eat in the home cage. FST was performed as described above on the subsequent day.

Sucrose preference test was performed as a two-bottle choice test over three consecutive days prior to sacrifice for both CSDS and CVS. One bottle was filled with water and the other bottle was filled with 2% sucrose. Initial weights of each bottle were recorded and bottle weights were recorded each morning and evening over the sucrose preference period. Sucrose preference was calculated as change in weight of the sucrose bottle/change in weight of both bottles × 100. Total sucrose preference over the 3-day period was used for analysis.

In order to allow for peak AAV expression at the time of behavior, viral infusions and behavior were separated by nearly 4 weeks as described in reported experimental timelines. In order to examine the effects of different AAV infusions on body weight, mice were weighed prior to infusion and then again at 3.5 weeks to reflect the effect of virus alone. Because male and female behavior took place at different times, comparisons between male and female mice on specific behavioral tests were not performed. However, sex differences in stress-dependent behaviors are well documented in the literature[15,42].

**Statistics**. Statistics were performed in Prism version 5.0 for Mac (GraphPad Software, La Jolla, CA, USA) and SPSS Statistics version 22 (IBM Corp, Armonk NY, USA). In cases where data met the assumptions necessary for parametric statistics, data from two groups were analyzed using a two-tailed Student's $t$ test and data from three or more groups were analyzed with a one-way analysis of variance (ANOVA). In cases where the data did not meet assumptions for parametric statistics, such as statistically different variance as determined by a Levene's test, data from two groups were analyzed using an independent samples Mann–Whitney and data from three or more groups were analyzed with a Kruskal–Wallis test. Post hoc statistical testing was performed when the $P$ value of the initial test rounded to a value that was ≤0.05. For parametric statistics, Tukey's and Bonferroni tests were used. For nonparametric statistics, different groups were compared by independent samples Mann–Whitney tests. In cases when the same

experimental test was performed different times with different settings (i.e., novel arena, home arena in NSF) and parametric statistics were permitted, a mixed-model ANOVA was employed. A mixed-model ANOVA concurrently analyzes the effect of one between-subjects measure (i.e., virus) and one within-subjects measure (i.e., arena). To assess differences in estrous cycle phase, a $\chi^2$ test was used and a linear regression test was employed to determine the association between SI and nuclear ERα expression. Outlier detection was performed for all sets using a Grubbs test with an α-value of 0.05. Statistical outliers were excluded from analysis. Statistical test utilized is reported in figure legends.

**Factor analysis**. In order to establish baseline effects of stress in females, we utilized factor analysis on the sum of all behavioral outputs. Since female behaviors in CVS stressed mice and unstressed controls did not occur on the same day, examining stress effects in individual tests was statistically inappropriate. However, by reducing the dimensionality of the independent behavioral variables with exploratory factor analysis, we were able to account for variability in the data due to differences in cohort. Standard factor analysis was performed using the scikit-learn package. We used 10-fold cross-validation (CV) to choose the number of actors and found that a single factor maximizes the CV log-likelihood (Supplementary Fig. 6a, b).

**Tissue collection**. Mice were killed directly from the home cage 24 h after final behavior. Brains were removed and cooled prior to slicing on a pre-defined brain matrix. Tissue was dissected and frozen immediately on dry ice. Two punches (bilateral) from each mouse were combined, but samples were not pooled between mice.

**Subcellular fractionation and Western blotting**. Frozen NAc tissue punches were fractionated into nuclear and cytosolic components using established procedures[43]. Two NAc punches were pooled bilaterally from a single animal and homogenized in 60 μL of 4-(2-hydroxyethyl)-1-piperazineethanesulfonic acid (HEPES)-buffered sucrose using a Teflon homogenizer at 400 rpm. Each sample was homogenized with 30 up and down strokes followed by centrifugation (1000 x g) at 4 °C for 10 min. Supernatant was collected as the cytosolic fraction and the resulting pellet was re-suspended in HEPES-buffered sucrose followed by additional centrifugation (1000 x g at 4 °C for 10 min). This procedure was repeated to yield a washed nuclear fraction, which was re-suspended in radio-immunoprecipitation assay buffer for analysis. Prior to protein quantification, samples were quantified using a DC protein assay (Bio-Rad). For each analysis, equivalent input protein concentrations were used. Lysates were calibrated to 12 μL of solution and 3 μL of reducing sample buffer containing sodium dodecyl sulfate (SDS) and dithiothreitol was added. Samples were mixed and heated at 95 °C for 5 min. Samples were removed from heat and centrifuged for 30 s before loading on a Criterion 4–15% Tris-HCl 1.0 mm precast gel (Bio-Rad). Proteins were separated based on molecular weight with SDS-polyacrylamide gel electrophoresis followed by membrane transfer to Immoblion-P 0.45 um pore side membranes (Millipore) at 100 V for one hour. Protein transfer was completed in SDS-free transfer buffer containing 15% methanol at 4 °C. To reduce non-specific binding, membranes were blocked in Tris-buffered saline containing 5% bovine serum albumin and 0.1% Tween-20 for 1 h at room temperature. Subsequently, membranes were incubated in primary antibody dissolved in blocking solution overnight at 4 °C. Following multiple washes, membranes were incubated in secondary antibodies containing peroxidase (Vector Labs) dissolved in blocking solution at room temperature for 2 h. Membranes were washed again, and then developed with chemiluminescent substrate (Thermo Scientific). Primary antibodies were used at a 1:1000 dilution and secondary antibodies were used at a 1:50 000 dilution. The following antibodies were used: ERα (Millipore, Cat# 06-935), ERβ (Abcam, Cat# ab3577), and aromatase (Cell Signaling, Cat# 14528). Protein expression was quantified using Image J software (United States National Institute of Health) using the densitometry tool to measure an equally sized rectangular space surrounding the protein on converted 8-bit images. In order to compensate for any irregularities in the gel or imaging, the space immediately above the protein of interest was measured as well and its intensity was subtracted as a control. In order to merge multiple cohorts, the control condition for each blot was normalized to an expression value of one. To compare protein expression between sexes, four male replicates were repeated on the female Western blots, and expression values were scaled accordingly. Full western blots for all samples used in analysis are shown in Supplementary Figs. 7–9. Blots were stripped and re-probed to analyze different proteins in the same sample.

**Viral reagents**. FLAG-tagged AAV expressing ESR1 was prepared by cloning a plasmid containing human ESR1 (Origene, SC125287) into the pCMV5-XL5 plasmid. A FLAG tag was added and the plasmid was packaged into AAV serotype 1 with a ubiquitous promoter chicken beta7 with a titer of $5 \times 10^{12}$ by the Vector Core of the University of Pennsylvania. AAV-GFP (AAV1.Cb7.Ci.eGFP.WPRE.rBG) was purchased directly from the Vector Core (Catalog Number: AV-1-PV1963) for use as a control.

**Viral-mediated gene transfer**. Stereotaxic surgeries targeting the NAc and PFC were performed as previously described[16]. Mice were anesthetized with an intra-peritoneal injection of ketamine (100 mg kg$^{-1}$) and xylazine (10 mg kg$^{-1}$) dissolved in sterile water. Subsequently, mice were placed in a small-animal stereotaxic device (Kopf Instruments) and the skull surface was exposed. 33-gauge needles (Hamilton) were utilized to infuse 0.5 μL of virus at a rate of 0.1 μL min$^{-1}$ followed by a 5-minute rest period to prevent backflow. The following coordinates were utilized for the NAc (from bregma: anterior/posterior +1.6 mm, medial/lateral +1.5 mm, dorsal/ventral −4.4 mm; 10° angle) and PFC (from bregma: anterior/posterior +1.8 mm, medial/lateral +0.75 mm, dorsal/ventral −2.7 mm; 15° angle). Due to viral spread, NAc injections were non-specific to shell vs. core. While PFC injections targeted the infralimbic cortex, virus spread sometimes extended beyond these anatomical boundaries to other regions of the PFC. Because AAV-ESR1 was FLAG-tagged and did not contain GFP, dissection of virally infected tissue for both AAV-GFP and AAV-ESR1 was performed based solely on anatomical boundaries (described above). However, because the ESR1 over-expressed (human) was not endogenous to the mouse, successful overexpression was confirmed following sequencing by aligning the reads to the human ESR1 gene. AAV-ESR1 injected mice that did not show up-regulation of human ESR1 were excluded from downstream analysis.

**Immunohistochemistry**. Mice were anesthetized with an intraperitoneal injection of ketamine (100 mg kg$^{-1}$) and xylazine (10 mg kg$^{-1}$) dissolved in sterile water and transcardially perfused with 20 mL 1× phosphate-buffered saline (PBS) over 5 min. Twenty microliters of 4 °C 4% paraformaldehyde (PFA) was injected over 5 min to fix the tissue. Brains were extracted and stored in 4 °C PFA overnight, followed by a 30% sucrose solution for additional storage. Brains were sectioned at 50 μm and washed in 1x PBS. Sections were blocked in a 1.5% bovine serum albumin solution (dissolved in PBS-Triton) at room temperature for 1 h, followed by incubation in a solution of primary antibody (in PBS-Triton) overnight at 4 °C. After a wash in PBS-Triton, sections were incubated in secondary antibody (in PBS-Triton) for 1 h at room temperature. Sections were washed a final time before being mounted on gelatin-coated slides and dried overnight. Vectashield hard antifade mounting medium with 4',6-diamidino-2-phenylindole (DAPI) (Vector Labs) was applied and the coverslip was added. All slides were analyzed using an LSM710 laser-scanning confocal microscope (Carl Zeiss). Placement and expression of virus was imaged in C57BL/6J mice using a x10 objective of 30 individual tiles in a single plane. Studies examining D1 and D2 specificity were performed on transgenic D1-GFP and D2-Tomato mice on a C57BL/6J background. In order to visualize neurons at a higher resolution, a x63 oil objective was utilized. In addition, we performed a Z-stack projection over several planes to maximize covered area. For each image, the center and borders were defined manually in real time, with a pre-defined number of images specified between the boundaries. The following anti-bodies were used: ERα (Millipore, Cat# 06-395), FLAG (Sigma, Cat# F1804), and GFP (Aves, Cat# 1020). Due to high florescence at baseline, tomato was not amplified prior to imaging.

**RNAscope in situ hybridization**. Fluorescent in situ hybridization (FISH) for Drd1 mRNA (Probe #406491), Drd2 mRNA (Probe #406501), and Esr1 mRNA (Probe #310301) was performed using the RNAscope® Fluorescent Multiplex 2.0 assay as per the manufacturer's instructions (Advanced Cell Diagnostics, Hayward, CA, USA)[44]. Briefly, fresh whole mouse brains were embedded in OCT medium and quickly frozen in isopentane (2-methylbutane) chilled to −80 °C. Twenty micrometer cryosections of NAc were then prepared and mounted on SuperFrost Plus slides. Sections were fixed and pre-treated according to the RNAscope® guide for fresh frozen tissue. After pre-treatment, sections were hybridized with Drd1, Drd2, and Esr1 FISH probes using the HybEZ Hybridization System. After several amplification sets, the sections were counterstained with DAPI and mounted using Prolong Gold. Immunoreactive cells were analyzed bilaterally in the NAc. Confocal images were acquired on a LSM710 confocal microscope (Carl Zeiss) using a x63 oil immersion objective.

**RNA isolation and sequencing**. Total RNA was isolated from frozen dissected NAc tissue using QIAzol lysis reagent and purified using the RNAeasy Mini Kit (Qiagen). RNA integrity was assayed using an Aligent 2100 Bioanalyzer (Aligent, Santa Clara CA, USA). Average RNA integrity number (RIN) values were above nine and samples with RIN values <8 were excluded from analysis. Each sample consisted of bilateral NAc punches from the same animal with no pooling between animals. Sample sequencing replications for male ESR1, male GFP, female ESR1, and female GFP-overexperessing mice were 6, 8, 6, and 7 respectively. Libraries were prepared using the TruSeq RNA Sample Prep Kit v2 (Illumina, San Diego CA, USA). Briefly, mRNA was polyA selected from the total RNA pool. Targeted mRNA was then fragmented and converted to complementary DNA (cDNA) with reverse transcriptase followed by cDNA size selection and purification with AMPure XP beads (Beckman Coulter, Brea, CA, USA). To identify each sample, strand-specific adapters were ligated to adenylated 3′ ends and an additional size selection step was performed. The cDNA library was then amplified using PCR. During this step, barcodes of 6 bp were added to the adaptors. Library quality and concentration was measured by the Bioanalyzer before sequencing. Libraries were

sequenced by the Genomics Core Facility of the Icahn School of Medicine at Mount Sinai using an Illumina HiSeq 2500 System with v3 chemistry and 100 bp single-end reads. Multiplexing was performed to ensure reads of 20 million for each sample.

**RNA-sequencing analysis**. Quality assessment of RNA-seq data revealed a >80% mapping rate. In order to maximize potential overlap with previously acquired data, we analyzed our sequencing data in a manner consistent with previous studies[15,16]. Male mice were analyzed as per Bagot et al.[2], with alignment to the mm9 reference transcriptome performed in Tophat[45], read count normalization and estimates of gene expression performed in cufflinks[46], and filtering for long noncoding genes and pairwise differential expression analysis performed in cuffdiff[46]. Data from female mice were analyzed as per Hodes et al. [5] whereby the voom[47] and limma[48] packages were utilized to generate differential expression signatures. For both sexes, differential expression comparisons were generated to compare the effects of ESR1 overexpression in the NAc (AAV-ESR1) to AAV-GFP in unstressed controls. Uncorrected signed $P$ values were utilized to generate a ranked list for RRHO, and subsets of the RRHO analysis were utilized for GO.

**Rank-rank hypergeometric overlap**. We utilized RRHO to probe the similarities and differences in transcriptional signatures across conditions. RRHO is a threshold-free test that can be applied to determine the extent and significance of overlap between two differential expression analyses[20]. This test is advantageous in that both directionality and magnitude of gene alterations are considered and has been used in an analogous way to identify similarities in transcriptional signatures across different brain regions in a single experiment[16]. RRHO was performed by generating signed-ranked lists of differential expression changes for each sequencing comparison by multiplying the −log 10($P$ value) by the sign of the fold change. In order to identify overlap between ESR1-dependent differential expression and stress-induced transcriptional profiles, we used previously validated R scripts[20,49] to perform a one-sided version of the RRHO test assessing enrichment only. RRHO difference maps were produced for pairwise comparisons comparing male ERα overexpression to the resilient phenotype, susceptible phenotype, and female ERα overexpression, and female ERα to 6-day female CVS. These maps calculate the difference in log odds ratio and standard error of overlap for each pixel in the pairwise comparison, which is converted to a $z$-score that is then corrected for multiple comparisons.

**Gene ontology**. We used the Panther GO[50] (www.pantherdb.org) to examine enrichment or statistical underrepresentation in our ranked differential expression lists. GO terms with a corrected statistical threshold of $P < 0.05$ were determined to be significant.

**Data availability**. The authors declare that the data supporting the findings of this study are available within the paper and its Supplementary Information files. Additionally, sequencing data for AAV-ESR1 vs. AAV-GFP comparisons in male and female mice are archived in the Gene Expression Omnibus accession number: GSE110725.

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

## Acknowledgements

This work was supported by National Institutes of Health grants F30MH110073 (Z.S.L.), T32GM007280 (Z.S.L.), T32MH096678 (Z.S.L.), and P50MH096890 (E.J.N.), NARSAD Young Investigator Award 22713 (R.C.B.) and the Hope for Depression Research Foundation. We thank Catherine J. Peña (Icahn School of Medicine at Mount Sinai) and Erin S. Calipari (Vanderbilt University) for their helpful comments on the manuscript.

## Author contributions

Z.S.L. and R.C.B. performed upstream regulator analysis. Z.S.L. and M.E.C. performed western blotting and Z.S.L., M.S., E.M.P., and H.K. performed IHC experiments. E.M.P. performed the RNAscope assay. Z.S.L., R.C.B., P.J.H., and B.L. performed stereotaxic surgeries and Z.S.L., R.C.B., P.J.H., D.M.W. and O.I. performed tissue dissections. Z.S.L. ran the behavioral testing and A.E.S. and M.Z. assisted with behavioral data analysis. Z.S. L., R.C.B., and D.M.W. performed the RNA-sequencing experiments and RNA-sequencing data was analyzed by L.S., Y.-H.E.L., and I.P. with further bioinformatics analysis performed by Z.S.L. and Y.E.L. G.E.H., M.L.P., and S.J.R. provided tissue from stressed female mice for protein analysis and T.Y.Z. and M.J.M. created and supplied the AAV-ESR1 virus. Z.S.L., R.C.B., and E.J.N. conceived experiments and interpreted data. Z.S.L. and E.J.N. wrote the manuscript. All authors edited the manuscript.

## Additional information

**Competing interests:** The authors declare no competing interests.

