## [Peer Review File(PDF 402 kb) · Nature Communications]

Reviewers' comments:

Reviewer #2 (Remarks to the Author):

This study uses a previously reported gene expression data set to identify key upstream regulator(s) of genes that are differentially expressed in the NAc of mice that are resilient to chronic social defeat stress. The results show that ER α is the top predicted upstream regulator in male mice that are resilient to CSDS compared to controls or to susceptible mice. The paper goes on to show that ER α protein levels are increased in nuclear fractions of the NAc of resilient male mice compared to susceptible mice (i.e., decreased in susceptible mice), and that exposing female mice to chronic variable stress (CVS) also decreases levels of ER α in nuclear fractions of NAc. The functional impact of increased ER α is tested by viral expression of ER α in the NAc of both male and female mice, and demonstrates that ER α is sufficient to increase resilience and produce antidepressant actions in both genders. Moreover, AAV- ER α overexpression is sufficient to reproduce the pattern of up-regulated genes observed in mice that are resilient to CSDS, although this effect was not observed in female mice demonstrating sex specific effects of ER α on gene expression. These results represent an interesting and important unbiased approach to identifying key upstream regulators involved in stress resilience. There are several points to address.

1. It would be interesting to test the effects of CVS in male mice to determine if similar or different upstream regulators are observed when males are exposed to a different stress paradigm.
2. It would also be interesting to know if ER α is regulated in other brain regions or if this effect is specific to the NAc?
3. The results demonstrate that AAV-ER α overexpression in the NAc is sufficient to produce a resilient behavior in both male and female mice. Related to the above regional specificity question, it would be interesting to test if the AAV- ER α effects are specific to the NAc.
4. Did AAV-ER α overexpression have any effects on locomotor activity or body weight?
5. For the studies to examine the gene expression profile resulting from AAV-ER α , please describe the dissection used to isolate AAV infected NAc. Also, what percent of the D1 and D2 cells in this dissection are virally infected? It would be interesting in future studies to examine cell specific gene expression profiles resulting from ER α expression. Finally, does the viral infusion target both shell and core?

Reviewer #3 (Remarks to the Author):

The authors provide interesting evidence of estrogen receptor alpha playing a pro-resilient role in the effects of chronic stress in both female and male mice. Their data show that ER α expression in the NAc is elevated in the mice resilient to the effect of chronic stress and the AAV-mediated overexpression of ER α in the NAc promoted stress resilience in both sexes.

They go on to characterize ERα mediated transcriptional changes in the NAc of resilient mice to conclude that ERα exerts a transcriptional response similar to stress resilience, but only in male mice. Overall, the findings are interesting and the manuscript in general is well-written. One or two introductory sentences of why the authors chose the NAc for IPA in the introduction would be helpful. Additionally, I have major concerns regarding the behavioral data included and its interpretation in the manuscript that I suggest the manuscript to be revised addressing the following questions and concerns.

1) My biggest concern in the manuscript is the behavioral testing and data presentation of the ERα overexpression in the NAc behavioral data.

A-There is not sufficient information on the behavioral testing for SI, OFT, FST, NSF and sucrose preference. References are included but major details are omitted that it is difficult to understand what the behavioral measurements included in figure 3 represent. For example, no target vs target means when the aggressor they experienced defeat was the target?

B-Were SI, OFT, and FST performed all in the same day for CSDS male mice and why? If so, there are major concerns of the behavioral data since each test can affect another the results of the other test.

C- Was the immobility duration in Figure 3 for male and female statistically different?

D- For proper interpretation of the pro-resilient effects of overexpressing ERα in the NAc, results presented in figure 3 and supplemental figure 3 should be combined. It would be more clear if the authors showed successful chronic stress effects in AAV-GFP mice, and then demonstrated how ERα overexpression results differed. Currently, it appears that not all behavioral measurements looked to be altered by CSDS or CVS, except SI. If the stress model did not clearly show behavioral changes, then the significant behavioral modifications described in the results/discussion sections between stressed AAV.GFP and AAV.ESR1 are limited in interpretation. The authors will need to revise their conclusions that ERα in NAc drives resilience to CSDS or CVS if they cannot show that in actuality it functionally increased resilience.

2) The authors mentioned that ESR1 is differentially expressed between resilient vs control and susceptible vs resilient mice (line 68-69). Would it have been expected that CSDS resilient male mice express more ERα in the NAc nuclear fraction compared to control in addition to susceptible mice?

3) In CSDS, authors were able to group male mice as resilient or susceptible based on SI performance. Why were female mice that underwent CVS not grouped in a similar method? Could the authors have used NSF or sucrose preference results to define mice as either resilient or susceptible? Would ERα levels differ between the 3 groups similarly to CSDS males?

4) In lines 87-90, the authors examined ERα labeling in D1 and D2 expressing cells. There is not a clear explanation of why they did this or how this contributes to the main aim of the manuscript or the particular results sections/figure it is placed.

5) The authors showed baseline sex differences in ERα, ERβ, and aromatase expression in

the NAc in supplemental figure 2. Could baseline sex-differences have contributed to sex-specific ER α transcriptional responses?

6) In Figure 3c and S3a, is a two-way repeated ANOVA an appropriate stat to use here?

Reviewer #2 (Remarks to the Author):

This study uses a previously reported gene expression data set to identify key upstream regulator(s) of genes that are differentially expressed in the NAc of mice that are resilient to chronic social defeat stress. The results show that ER α is the top predicted upstream regulator in male mice that are resilient to CSDS compared to controls or to susceptible mice. The paper goes on to show that ER α protein levels are increased in nuclear fractions of the NAc of resilient male mice compared to susceptible mice (i.e., decreased in susceptible mice), and that exposing female mice to chronic variable stress (CVS) also decreases levels of ER α in nuclear fractions of NAc. The functional impact of increased ER α is tested by viral expression of ER α in the NAc of both male and female mice, and demonstrates that ER α is sufficient to increase resilience and produce antidepressant actions in both genders. Moreover, AAV- ER α overexpression is sufficient to reproduce the pattern of up-regulated genes observed in mice that are resilient to CSDS, although this effect was not observed in female mice demonstrating sex specific effects of ER α on gene expression. These results represent an interesting and important unbiased approach to identifying key upstream regulators involved in stress resilience. There are several points to address.

1. It would be interesting to test the effects of CVS in male mice to determine if similar or different upstream regulators are observed when males are exposed to a different stress paradigm.

This is a great thought. In order to address this, we looked at upstream regulators in the CVS model that we utilize later in the paper. We did this for both male and female mice (RNA sequencing data from Hodes et al. 2015 J Neuroscience) and include these data as supplemental figure 1B. Interestingly, ESR1 is a predicted upstream regulator (predicted downregulation) in the NAc for male, but not female, CVS, while this paradigm alters behavior in female, but not male, mice. These findings coincide with our CSDS data (figure 1B) in which we only see predicted upregulation for ESR1 in resilient male mice. Since CVS does not produce a resilient subset, we would not expect to see an upregulation of ESR1 and therefore these data support our proposed role for ESR1 as a resilient-specific molecular adaptation.

2. It would also be interesting to know if ER α is regulated in other brain regions or if this effect is specific to the NAc?

As with above, we addressed this by performing upstream regulator analysis on RNA sequencing data, though this time, we utilized the Bagot et al. 2016 Neuron CSDS dataset that gave us our original ESR1 NAc prediction (figure 1B). These new data have been included as supplemental figure 1A and show no predicted upregulation for ESR1 in the prefrontal cortex, a brain region with connections to the NAc that is also implicated in MDD.

3. The results demonstrate that AAV-ER α overexpression in the NAc is sufficient to produce a resilient behavior in both male and female mice. Related to the above regional specificity question, it would be interesting to test if the AAV- ER α effects are specific to the NAc.

In order to confirm our bioinformatics prediction that AAV-ESR1 would have no effect in the PFC, we overexpressed AAV-ESR1 or AAV-GFP in male mice and exposed them to

CSDS. AAV-ESR1 in the PFC did not have an effect on social interaction time (now included as supplemental figure 5c) or sucrose preference (now included as supplemental figure 5d), confirming our bioinformatics prediction of regional specificity for the pro-resilient effect of AAV-ESR1 in NAc.

4. Did AAV-ER α overexpression have any effects on locomotor activity or body weight?

Given the nature of our behavioral data this is an important control to include in the manuscript and is now featured in supplemental figure 5a-b. These data show that AAV-ESR1 did not have an effect on either locomotor activity or body weight.

5. For the studies to examine the gene expression profile resulting from AAV-ER α , please describe the dissection used to isolate AAV infected NAc. Also, what percent of the D1 and D2 cells in this dissection are virally infected? It would be interesting in future studies to examine cell specific gene expression profiles resulting from ER α expression. Finally, does the viral infusion target both shell and core?

We now include new RNAscope data showing that AAV-ESR1 infects roughly equal numbers of D1 and D2 cells in the NAc (see supplemental figure 4). This is expected since we have shown previously that AAV vectors do not distinguish between subtypes of neurons within infected brain regions, nevertheless, it is a useful control and we appreciate the reviewer's prompt to generate these data. This finding also establishes the physiological nature of the overexpression study, since we already show in the manuscript that ER α is expressed at roughly comparable levels in D1 and D2 cells. As the reviewer requests, we also now include details of the dissection for AAV-infected NAc in the Methods. Because the virally-expressed ESR1 is FLAG-tagged, we were able to confirm transgene overexpression in our sequencing data. As the mouse NAc is small, our viral infusions encompass both shell vs. core, a point now made explicitly in the Methods. Finally, we definitely agree with the reviewer that cell specific gene expression profiles would be a very interesting future direction, but as the reviewer points out, this is well beyond the scope of the current manuscript.

Reviewer #3 (Remarks to the Author):

The authors provide interesting evidence of estrogen receptor alpha playing a pro-resilient role in the effects of chronic stress in both female and male mice. Their data show that ER α expression in the NAc is elevated in the mice resilient to the effect of chronic stress and the AAV-mediated overexpression of ER α in the NAc promoted stress resilience in both sexes. They go on to characterize ER α mediated transcriptional changes in the NAc of resilient mice to conclude that ER α exerts a transcriptional response similar to stress resilience, but only in male mice. Overall, the findings are interesting and the manuscript in general is well-written. One or two introductory sentences of why the authors chose the NAc for IPA in the introduction would be helpful. Additionally, I have major concerns regarding the behavioral data included and its interpretation in the manuscript that I suggest the manuscript to be revised addressing the following questions and concerns.

We would like to thank the reviewer for his/her feedback. We are pleased that they found the study interesting. In order to address these general comments, we have added

introductory sentences on our selection of brain regions, which now includes both the NAc and PFC.

1) My biggest concern in the manuscript is the behavioral testing and data presentation of the ERA overexpression in the NAc behavioral data.

A-There is not sufficient information on the behavioral testing for SI, OFT, FST, NSF and sucrose preference. References are included but major details are omitted that it is difficult to understand what the behavioral measurements included in figure 3 represent. For example, no target vs target means when the aggressor they experienced defeat was the target?

We apologize for the lack of detail in our original manuscript and very much appreciate the reviewer pointing out this need. We take this comment very seriously as we wish to maximize reproducibility of our experiments in other laboratories. As a result, we have greatly expanded our Methods section to include detailed information on behavioral testing for SI, OFT, FST, NSF, and sucrose preference. Included in this is a discussion of what is meant by “no target” and “target”, which we have changed to “absent” and “present” since we believe this adds further clarity for the reader.

B-Were SI, OFT, and FST performed all in the same day for CSDS male mice and why? If so, there are major concerns of the behavioral data since each test can affect another the results of the other test.

As shown in the timeline in figure 3a SI, OFT, and FST were performed in the same day. We agree with the reviewer that behavioral data from one test has the potential to affect the results of another test. However, in practice, we rarely see this and believe that in our case the benefits of performing the testing on the same day greatly outweighed the disadvantages. Importantly, even if there are effects of one test on a subsequent test, all mice are treated precisely the same way such that this would not confound our overall analyses. Therefore, the fact that all tests were performed on the same day does not detract from the validity of our data. An important consideration is that most of our group’s studies utilize HSV vectors, which express transgenes for 3-4 days only. Adhering to our “typical” behavioral timeline in the present study thus maximizes comparisons with all of our other published data, including Bagot et al. 2016 Neuron to which our data is explicitly compared. Moreover, we find that we can mitigate spillover effects by separating the tests by a minimum interval of two hours and performing the most stressful (FST) procedure at the end of the day, which we did in this case (this is now specified in the Methods along with a statement clarifying our timeline for SI, OFT, and FST and rationale for same-day testing). Further, the tests in which we saw clear significant effects (CSDS and sucrose preference) were the first and last tests, so it is unlikely that the tests in between affected the behavior of the groups. Critically, our “typical” timeline for behavioral testing, which includes behavioral tests in addition to SI on the same day has been used extensively in the published literature (for example, see Labonte et al. 2017 Nature Medicine, Bagot et al. 2016 Neuron, Sun et al. 2015 Nature Medicine). In conclusion, we appreciate the reviewer’s consideration of this rationale and believe that the consistency across all animals and comparability to published data provide a very strong rationale for our approach.

C- Was the immobility duration in Figure 3 for male and female statistically different?

Unfortunately, we cannot directly compare male and female data since they were run at different times. As such, statistical comparisons for sex differences in behavior would not be appropriate. However, sex differences in behavioral tests have been well established, though the directionality of these effects in FST conflict across datasets (see Korkas and Dalla 2014 British Journal of Pharmacology). We have now added a sentence in the Methods on our inability to statistically analyze male/female behavioral comparisons despite known sex differences. The important conclusion from our data, and the rationale to include both sexes in our study, was to demonstrate whether ESR1 exerts pro-resilient effects in males and females, something we have achieved.

D- For proper interpretation of the pro-resilient effects of overexpressing ER α in the NAc, results presented in figure 3 and supplemental figure 3 should be combined. It would be more clear if the authors showed successful chronic stress effects in AAV-GFP mice, and then demonstrated how ER α overexpression results differed. Currently, it appears that not all behavioral measurements looked to be altered by CSDS or CVS, except SI. If the stress model did not clearly show behavioral changes, then the significant behavioral modifications described in the results/discussion sections between stressed AAV.GFP and AAV.ESR1 are limited in interpretation. The authors will need to revise their conclusions that ER α in NAc drives resilience to CSDS or CVS if they cannot show that in actuality it functionally increased resilience.

This is a very good point. In order to better exhibit the differences between unstressed and stressed mice we have combined the control and stressed data for male CSDS as requested (figure 3). The statistics now represent the effects of CSDS as a whole in addition to the effects of virus and the reader can visually compare all groups. Unfortunately, for logistical reasons, it was impossible for us to run the female control and stressed cohorts on the same day. As such, the two groups were staggered by a day. Given this, we believe that it would be statistically inappropriate to compare the groups directly and have opted to keep the groups separate (figure 4 and supplemental figure 6). For transparency purposes, we have now included a statement that female control and stressed cohorts were not run in parallel in the Methods section. However, as the reviewer points out, it is very important to establish baseline stress effects before concluding about a stress-specific response. In order to address this for our female cohort, we performed factor analysis on the totality of behavioral outputs (which eliminates the variable of time). This is now featured in figure 4b and shows a significant virus by stress interaction for factor 1, the single factor that gives us the maximum likelihood in our analysis. Given this and the SI data in figure 3c we believe it is reasonable to conclude that our stress significantly affected behavior in both males and females and therefore, our conclusion that AAV-ESR1 increases behavioral resilience is valid.

2) The authors mentioned that ESR1 is differentially expressed between resilient vs control and susceptible vs resilient mice (line 68-69). Would it have been expected that CSDS resilient male mice express more ER α in the NAc nuclear fraction compared to control in addition to susceptible mice?

Not necessarily. As shown in figure 2a, expression of nuclear ER α in the NAc is a gradient from lowest (susceptible) to middle (control) to highest (resilient) that is significant by one-way ANOVA, a finding that we further clarify with our correlation analysis in supplemental figure 2c. We interpret this as deviation from baseline (control) levels, whereby the directionality of ER α change in the NAc is dependent on the

individual mouse's response to stress. If the mouse is resilient, we see an increase from baseline. If the mouse is susceptible, we see a decrease from baseline. As such, we are predominantly concerned with the difference between susceptible and resilient mice, which is statistically significant. In order to make this clearer to the reader, we have included a discussion of this topic in our Discussion section.

3) In CSDS, authors were able to group male mice as resilient or susceptible based on SI performance. Why were female mice that underwent CVS not grouped in a similar method? Could the authors have used NSF or sucrose preference results to define mice as either resilient or susceptible? Would ER α levels differ between the 3 groups similarly to CSDS males?

This is a fantastic idea, and we would love to be able to examine levels of ER α in female mice to see if we get a similar relationship to what we show in figure 2a. Unfortunately, in contrast to CSDS, there are no clearly established cutoff criteria for what constitutes "susceptibility" or "resilience" to CVS. The reason for this is that CSDS produces a bimodal distribution in the social interaction test (see Krishnan et al., 2007 Cell) that is not present in the behavioral outputs of CVS. However, the reviewer's suggestion is a great one, and we hypothesized that the combination of behaviors might be powerful enough to identify a clear resilient population, so we tried to address this by using our factor analysis. Unfortunately, despite showing a significant stress x virus interaction overall ($p < 0.05$), our analysis still shows a normal unimodal distribution in our GFP stressed and GFP control mice, which were the only groups where we could analyze ER α expression without interference from AAV-ESR1. As such, we feel that we are unable to determine exactly what constitutes "resilience" in this context and cannot determine whether ER α levels vary between categorical groups similar to males. We do see, however, that stress is associated with a significant reduction in ER α protein in the cell nucleus (figure 2c), but we are unable to extend this finding to speculate on levels of ER α in CVS resilient mice.

4) In lines 87-90, the authors examined ER α labeling in D1 and D2 expressing cells. There is not a clear explanation of why they did this or how this contributes to the main aim of the manuscript or the particular results sections/figure it is placed.

We would like to thank the reviewer for pointing this out. We examined ER α expression at baseline in D1 and D2 expressing cells because the NAc is a heterogeneous tissue and we wanted to see if either of these two principal neuron subtypes was more likely to be driving our responses, since this would inform on the utility of future cell-type specific approaches. While we cannot rule out that one specific cell type is driving our behavioral or transcriptional responses from the current data, and hope to explore this in our future studies, our findings that D1 and D2 expressing cells express ER α equally (figure 2g), and that AAV-ESR1 has similar tropism for D1 and D2 expressing neurons (new RNAscope data: supplemental figure 4), make this less likely. However, in order to address the reviewer's question, we have added to our justification for the D1 and D2 IHC experiment in both the Results and Discussion sections.

5) The authors showed baseline sex differences in ER α , ER β , and aromatase expression in the NAc in supplemental figure 2. Could baseline sex-differences have contributed to sex-specific ER α transcriptional responses?

Absolutely. This is a great observation and something we have been talking about internally. We have added this point to the Discussion section.

6) In Figure 3c and S3a, is a two-way repeated ANOVA an appropriate stat to use here?

We apologize for the confusion. The analysis we were performing in both figure 3c and figure S3a is an ANOVA that examines the effect of two factors (virus and presence/absence of target), one of which (time in interaction zone when the target is absent and present) has two values per mouse. In Prism 5, this is called a two-way repeated measures ANOVA. However, in most contexts, this is called a mixed-model ANOVA. To be more precise, we have changed the name of this analysis from two-way repeated measures ANOVA to mixed model ANOVA in the text when we use this analysis (now only in figure 4c), and clearly delineated the details of the analysis in the methods section. Thank you for catching this discrepancy!

REVIEWERS' COMMENTS:

Reviewer #2 (Remarks to the Author):

The authors have addressed all of the comments and concerns raised during the initial review of the manuscript, including the addition of new experiments and results.

Reviewer #3 (Remarks to the Author):

The authors have addressed all my criticisms and the revised manuscript is much improved. There are only a few more minor points that I believe need to be addressed to further enhance their manuscript.

- 1) Add "male" to at least the first sentence (lines 121-122) to clarify the sex of the mouse you are studying even if it says CSDS.
- 2) Harris et al published recently on a new method for chronic social defeat stress in female mice (Neuropsychopharmacology 2017,doi:10.1038/npp.2017.259), which the effects of CSDS in the females were similar to those found in males. It is understood that CSDS was used in males and CVS was used in females as chronic stress since CSDS was not possible in female mice, but these two forms of chronic stress can differ in effects. Could the authors briefly address whether utilization of this female model may be of relevance/importance to further understand their current findings in the future?
- 3) For future reproducible experiments, please include vendor information for the mice and state when the behavioral testing occurred (light or dark phase).
- 4) Minor points on figures. Figure 2G, instead of GFP, change to D2 even though stained for GFP where D1 was not stained. This could make it clearer. Figure 5e title says "Female SCVS: stress phenotype", but it is supposed to state Female CVS?

Reviewer #4 (Remarks to the Author):

In this revised manuscript, Nestler and colleagues include additional data and analyses to support their conclusion that Estrogen Receptor alpha could drive pro-resilient transcription in mouse models of depression. The authors have successfully addressed the concerns raised in the first round of review. The only concern that I have is that the authors did not demonstrate the direct regulation of altered gene expression in mouse models of depression by Estrogen Receptor Alpha as predicted bioinformatically. The data presented in the revised manuscript are correlative.

REVIEWERS' COMMENTS:

Reviewer #2 (Remarks to the Author):

The authors have addressed all of the comments and concerns raised during the initial review of the manuscript, including the addition of new experiments and results.

We are pleased that reviewer #2 found our revisions helpful for the overall quality of the manuscript.

Reviewer #3 (Remarks to the Author):

The authors have addressed all my criticisms and the revised manuscript is much improved. There are only a few more minor points that I believe need to be addressed to further enhance their manuscript.

1) Add "male" to at least the first sentence (lines 121-122) to clarify the sex of the mouse you are studying even if it says CSDS.

This has been added as requested.

2) Harris et al published recently on a new method for chronic social defeat stress in female mice (Neuropsychopharmacology 2017,doi:10.1038/npp.2017.259), which the effects of CSDS in the females were similar to those found in males. It is understood that CSDS was used in males and CVS was used in females as chronic stress since CSDS was not possible in female mice, but these two forms of chronic stress can differ in effects. Could the authors briefly address whether utilization of this female model may be of relevance/importance to further understand their current findings in the future?

A brief discussion of the two recent papers on female CSDS (Harris et al. and Takahashi et al.), and their role in future studies on ESR1 have been included in the discussion session. In particular, we state how use of female CSDS may be useful to identify a female resilient transcriptional profile which could then be compared to ESR1 overexpression.

3) For future reproducible experiments, please include vendor information for the mice and state when the behavioral testing occurred (light or dark phase).

Mice were tested during their light phase. This has been included in the methods.

4) Minor points on figures. Figure 2G, instead of GFP, change to D2 even though stained for GFP where D1 was not stained. This could make it clearer. Figure 5e title says "Female SCVS: stress phenotype", but it is supposed to state Female CVS?

These changes have been implemented.

Reviewer #4 (Remarks to the Author):

In this revised manuscript, Nestler and colleagues include additional data and analyses to support their conclusion that Estrogen Receptor alpha could drive pro-resilient transcription in mouse models of depression. The authors have successfully addressed the concerns raised in the first round of review. The only concern that I have is that the authors did not demonstrate the direct regulation of altered gene expression in mouse models of depression by Estrogen Receptor Alpha as predicted bioinformatically. The data presented in the revised manuscript are correlative.

While the editors have agreed to set this comment aside for the purpose of future revision, we would like to address it here textually. We disagree with the referee's assertion that we did not show direct regulation of estrogen receptor alpha in mouse models. We showed significant increases in nuclear estrogen receptor alpha in the NAc of male resilient mice following CSDS and a significant decrease in nuclear estrogen receptor alpha in this region of female mice following CVS. Since the cytosolic protein levels were unchanged, this is clearly the product of altered gene expression. Further, we overexpressed ESR1 and showed effects on gene expression, which is a direct confirmation of our bioinformatics predictions. Therefore, the idea that our data are "correlative" is incorrect.